# Challenges and Construction Applications of Solid Waste Management in Middle East Arab Countries

Osama Ibrahim *, Ghassan Al-Kindi, Mohsin Usman Qureshi and Salma Al Maghawry

Faculty of Engineering, Sohar University, P.O. Box 44, Sohar 311, Oman
* Correspondence: oibrahim@su.edu.om

**Abstract:** Over the past few decades, solid waste production, specifically construction waste, in Middle Eastern Arab countries has dramatically increased. This is characterized by several factors, including rapid urbanization, common food wasting habits, diverse culture, lack of proper planning of solid waste processes, insufficient equipment, as well as lack of proper funding. The exponential growth in solid waste generation rates has led to hazards to health and the environment, causing issues related to air and water pollution under the already increasing pressure of climate change. In this review, we analyze the current solid waste challenges in 13 Arab countries, common diseases, and actual projects applied. The selection of Arab countries was mainly based on the countries with the highest population as well as the availability of data in the field of study. This review also highlights the efforts of the Arab governments that implemented several pilot projects that are not sustainable or effective in the long term. We discuss the main issues facing each Arab country and the main challenges they have in common, as well as the potential to use the great amounts of construction waste in these countries. It is recommended that proper disposal and collection plans should be prioritized in the municipalities' agendas since air and water pollution represent the main challenge in all Arab countries. Adequate treatment, recycling, and compost production facilities should be initiated and monitored regularly to take advantage of the relatively high percentage of organic matter in most Arab countries. Proper cooperation between the informal sector, private companies, and governments should be ensured in order to achieve long-term goals in the solid waste management (SWM) sector in MENA (Middle East and Northern Africa) Arab countries. This review provides a comprehensive study of the construction waste in MENA Arab countries that will help reach the goal of achieving sustainable countries.

**Keywords:** municipal solid waste management; MENA; Arab countries; construction solid waste; sustainability; health hazards; construction applications

## 1. Introduction

Solid waste management has been an issue for centuries for both urban and rural areas in several countries. Despite its many effects on nature and human health, solid waste represents a valuable resource that could be effectively used to produce energy and create a financial return. Solid waste is any waste that is no longer useful to the first user. Solid waste management (SWM) is the process of treating and disposing of these materials. The source of solid waste is what determines its characteristics and properties as well as its composition. The composition of solid waste is the main factor controlling how it could be used in the future. Garbage and rubbish are the main types of solid waste; garbage describes compostable solid waste, construction, and demolition while rubbish is the dry materials and bulky items that cannot be decomposed. A significant part of SWM is recycling. Recycling includes material that is not part of trash or garbage [1].

The current solid waste trends show that by 2050, the global generation rate of solid waste is predicted to reach 27 billion tons per year, with Asia accounting for 30% of the generated waste. Within Asia, there are almost 760,000 tons of solid waste generated daily;

China generates almost 0.5 kg per capita per day and India generates 0.7 kg per capita per day [2]. Other estimations predict that Asian solid waste production is expected to rise to 1.8 million tons of solid waste generated per day in 2025. The dramatic increase in solid waste production rates can be due to several factors. These factors mainly include the rapid development of industrial zones, urbanization, and the massive increase in population. These factors lead to the actual numbers being almost twice as high as the estimated numbers.

Solid waste processes include collection, treatment, disposal, and recycling. These processes are found to be rare and insufficient in developing countries [3]. However, the amount of solid waste produced by developing countries is smaller than the amount produced by developed regions. In India, it was found that in 2001, the population was 1029 million, which is predicted to increase to 1400 million by 2028 [4]. It was also found that the highest percentage of the solid waste produced from these in India is domestic waste, which includes food waste, construction, and demolition waste, which is usually buried at construction sites [5]. The improper disposal of waste causes damage to infrastructure as well as contamination of water resources.

Egypt has the highest population among Arab countries in the Middle East region, followed by Iraq, Saudi Arabia, Yemen, Syria, Jordan, the United Arab Emirates, Lebanon, Palestine, Oman, Kuwait, Qatar, and Bahrain [6]. The most populated countries, which are Egypt and Iraq, have populations of 106 million and 42 million people, respectively. The least populated country, Bahrain, has a population of 1.7 million [7]. In all Middle Eastern Arab countries, SWM has been a constant concern due to its effects on SWM workers as well as the increase in the pollution of water and vegetation.

Processes such as treatment recycling and reusing by municipalities were found to be rare in Arab countries. Several studies have analyzed and reviewed the SWM status and statistics in Arab countries. Researchers show that Arab countries are responsible for millions of tons of solid waste annually. It was found that the countries that consume more natural resources usually produce a higher amount of waste [8]. These countries include Saudi Arabia, Oman, the United Arab Emirates, Kuwait, and Qatar. When analyzing the composition of solid waste produced by these countries, it was found that organic compostable waste represents 35% to 65% of the total solid waste produced [9]. The country with the highest amount of organics was found to be Yemen, indicating the potential of using this organic matter in compost production.

There is a lack of social awareness regarding SWM issues [10]. There is a common mentality that solid waste should be disposed of as soon as possible and not viewed as a source of energy that could be taken advantage of. Due to this mentality, several economic, health, and environmental risks have been faced by countries such as Lebanon in the past couple of years, creating a crisis in the country in which waste is disposed of randomly everywhere around the streets [11].

The disposal of solid waste represents the most challenging process of SWM processes in which proper disposal methods that take into consideration the environmental risks and health risks must be created and managed carefully [12].

Due to the current non-arranged systems of SWM, solid waste is disposed of at random dumps around cities and in streets. Moreover, the assigned containers were found to be of inadequate capacity, leading to waste being disposed around it rather than in it. This causes aesthetic issues as well as danger to public health and the infrastructure of a city. The collection, storage, treatment, transportation, and disposal processes of SWM must be organized properly in order to reduce the solid waste's negative impacts on the environment and take advantage of waste in a way that benefits the citizens and the government.

There is a noticeable lack of comprehensive SWM systems in Middle Eastern Arab countries. The complications in the SWM aspect are mainly caused by insufficient planning as well as the lack of laws and regulations that organize the solid waste disposal process. In order to achieve a successful SWM system, there should be organized cooperation

between governments, private companies, and educational institutions, as well as sufficient funds. When meeting these requirements, followed by proper techniques, equipment, and technologies, there could be a noticeable change in the SWM situation in Middle Eastern Arab countries.

The United Arab Emirates and Qatar are the most successful examples in terms of SWM in Middle Eastern Arab countries. There have been implementations of SWM programs in several regions of these two countries. Overall, governments and the private sector have made noticeable efforts in the processes of collection and transportation of solid waste as well as disposing of this waste in landfills that are properly designed based on engineering specifications. Regardless, the effectiveness of these efforts has not yet been confirmed, showing that more efforts and planning are required to reach the SWM goals in the region.

The need for these plans and efforts is urgent specifically in Middle Eastern Arab countries due to the rapid increase in solid waste generation rates throughout the region. The current efforts regarding increasing public awareness and slowly cultivating more developed techniques in the SWM sectors are not enough to lead to successful, noticeable results. In light of the aforementioned challenges, studying the current status of solid waste generation rates, composition, and the already implemented plans and efforts and their success is essential in order to reach the goals of successful SWM systems.

Successful SWM factors are key to achieving clean sustainable countries in the Arab world. One of the main key factors towards successful SWM is flexibility. Any SWM plan should have a margin of change in case of any expected changes in the economy, climate, political status, and people's habits, all of which affect the generation and characteristics of solid waste produced. Moreover, support from the main stakeholders in the country is essential to ensure sufficient resources for the development of an efficient SWM system. Another key factor is having clear objectives towards minimizing and material recovery, where the aim is to minimize the generated amount and recycle as much material as possible, especially that which can be used for construction purposes in a clean, environmentally friendly way. Finally, the success of any SWM plan greatly depends on the awareness of society, since public behavior is one of the main causes of the issue.

This paper provides a full review of the SWM challenges and issues in 13 Middle Eastern Arab countries, as well as the applied projects and construction applications.

This paper also provides recommendations and solutions for municipalities in order to protect the countries from the disadvantages of solid waste based on the main challenges found in each country and the main issues found in common for all studied countries, as well as point out the suitable construction applications for solid waste that may lead to a financial return in Middle Eastern Arab countries.

## 2. Challenges and Issues of Solid Waste in MENA Arab Countries

Rats and insects that move in landfills transmit diseases such as dysentery, diarrhea, Amoebic dysentery and plague, salmonellosis, trichinosis, endemic typhus, cholera, jaundice, hepatitis, gastrointestinal diseases, and malaria, without the reported cases of poisoning that may occur. Solid waste also affects the environment in many ways such as smells, and toxic gases are constantly emitted into the atmosphere and affect global warming and rainwater intrusion through the open dumping contamination of groundwater resources (leaching). A summary of the negative effects and challenges of solid waste follows.

### 2.1. Egypt

Several challenges face the SWM industry in Egypt. These challenges include the lack of highly skilled laborers that have the ability to use developed tools in order to minimize the amount of solid waste; the shortage of public land, which leads to solid waste being dumped in random uncontrolled areas, causing severe damage to the environment and people's health; the lack of budget dedicated to SWM solutions due to their high cost;

the lack of facilities and equipment that can be used for recycling; the lack of coordination between the government and the private sector; as well as the poor management in terms of construction activities at construction sites [13,14]. The random dumpsites and uncontrolled landfills lead to the leakage of leachate into the groundwater. This affects the groundwater quality and produces high concentrations of toxins and heavy metals that cause health diseases and major issues [15]. Studies have found that the improper disposal of hazardous waste had led to the increase in the endemicity of hepatitis B virus (HBV) infection, respiratory infections, gastrointestinal infections, and skin infections, especially among solid waste workers [16,17]. Using the high percentage of organic waste in highly populated regions, a suitable proposed solution is anaerobic digestion systems [18]. Moreover, the Egyptian Waste Management Regulatory Authority has been increasing recycling grades and increasing refuse-derived fuel production [7]. The political agenda of Egypt Vision 2030 includes several major projects such as the national project for the development of Siena, the national project of roads, and the establishment of a couple of new cities in which it is planned to include several recycling applications. Finally, in 2020, the waste management regulation law was issued by the Egyptian ministry of environment in hopes of controlling random disposing methods.

### 2.2. Iraq

The main challenges facing the Iraqi government are the side effects of ISIS damages and war happening in Iraq in recent years, financial constraints that prevent enough power from being produced to meet demand, and finally, the population growth rate [19]. The proposed solutions to mitigate these problems are as follows. The country must improve its domestic energy infrastructure, particularly in the power industry. The reusing system is performed by facilities in the following process: the waste is collected and then separated to be sold to industries to reuse it safely [20]. However, it was noticed that the disposal sites are not hygienic, and waste recycling is not common in the city; as a result, the recycling system seems to not gain as much popularity in Iraq's SWM system [21].

### 2.3. Saudi Arabia

There are many challenges and negative effects in SWM facing the KSA. First, as more than 75% of the country's population lives in cities, the government must take steps to improve the country's recycling and waste management situation [22]. Second, uncontrolled dumping of waste (in non-engineered landfills) has the potential to pollute groundwater and soil, as well as attract disease-carrying insects and rats. Next, the improper collection and management of leachate could pollute the soil and water. Then, a bad odor is created when solid waste heaps decompose biologically. Anaerobic conditions are created by the compaction of landfill layers and the biodegradability of organic waste, resulting in the creation of methane gas. The landfill gas might start a fire and cause a disaster [23]. There are different solutions to reduce the negative effects of solid waste. One is the conversion of waste such as food waste, manure, and plant residues to compost for agricultural purposes. Waste can be a significant source of valuable products and energy, in terms of waste composition and energy needs in Saudi Arabia. Technologies such as anaerobic digestion and pyrolysis have received a lot of attention. In addition, waste has economic benefits; for example, biogas can be utilized as a sustainable energy source that is cheaper than traditional fossil fuels because it contains up to 70% methane. The energy recovered from solid waste is environmentally safer because anaerobic digestion does not produce any more greenhouse gases. Finally, the improved biogas has a high probability of being used as a vehicle fuel or as a source of electricity for the grid. Saudi Arabia's energy requirement is 55,000 MW, which is provided by fossil fuels and natural gas. Upgrading biogas offers a way to minimize natural gas demand and fossil fuel dependence. Municipal solid waste in Saudi Arabia is collected from individual or community bins and disposed of in landfills or dumps. An active informal sector drives waste sorting and recycling. The recycling rate

ranges between 10% and 15%, owing to the presence of an informal sector that extracts paper, metals, and plastics from municipal waste.

### 2.4. Yemen

The main challenge that faces SWM in Yemen is the fact that there are currently no policies or laws for SWM [9]. As a result, waste is dumped randomly with no regulation. Moreover, dumpsite locations are not selected based on engineering parameters that include hydrological or topographical suitability. There is also no protection whatsoever for the soil beneath these non-sanitary landfills, leading to the contamination of soil and groundwater. Finally, there is a limited budget dedicated to the construction of engineering landfills [24]. The random disposal of solid waste leads to air pollution as well as groundwater pollution, and this problem becomes more dangerous in seasons of high rainfall intensity. AIDS (Acquired Immune Deficiency Syndrome), hepatitis, and tuberculosis are some of the diseases that were found to be common in the Yemeni community due to the improper disposal of solid waste [25]. Solutions that might be proposed to solve the increasing issue of SWM in Yemen include involving the private sector to increase the treatment projects, introducing laws that include public cleaning, preparing an awareness program and conferences, training the local people in order to increase the social awareness level regarding the issue [26].

### 2.5. Syria

The waste treatment industry is thought to be underdeveloped in Syria [27]. Due to the destruction of infrastructure, damage or looting of collection vehicles and waste containers, the devastation of government institutions, and the displacement of residents into safe areas as a result of the 2011 conflict, municipal SWM in Syria is a serious challenge for both national and local authorities. Although municipal or private organizations collect 85% of all solid waste in all Syrian towns and the majority of rural villages, an estimated 80% is disposed of at open dump sites on the outskirts of towns. This results in air pollution when dioxin and other pollutants are emitted during open-air burning, a common procedure for reducing waste volumes. Hazardous and non-hazardous wastes were usually mixed with home waste, posing a threat to water, soil, and air quality. Hazardous waste makes up roughly 1% to 3% of overall waste volume, but it was identified as one of Syria's most significant sources of pollution due to poor management. Waste segregation was not adopted at Syria's medical centers until 2010, posing a risk to healthcare employees, waste handlers, patients, and the general public.

### 2.6. Jordan

In Jordan, in the summer, the smell of rotten waste begins to spread, and the spread in the atmosphere of many gases resulting from waste causes many diseases. A large proportion of solid waste in Jordan can be recycled (65% of total waste). Only 5–7% is recycled or salvaged, mainly by the informal sector, and it faces some challenges, such as the high price of electricity, limited storage space, and exposure to price fluctuations; for glass, the internal demand is weak, so it is difficult to recycle in Jordan and there is no official recycling agency. The common method in Jordan for dealing with and disposing of the waste is landfills; there are 21 working landfill sites in Jordan, of which seven are closed landfill sites.

### 2.7. United Arab Emirates

The SWM industry in the UAE faces several challenges, including the inadequate quality of the solid waste produced, the poor design that leads to an excessive amount of solid waste, the huge budget required to initiate SWM projects, as well as the fact that some solid waste types are not suitable for all SWM processes [28–30]. Solid waste leads to emissions in the air and the pollution of soil, consequently leading to health effects that include diseases such as neurological diseases and cancer. Several solutions have been

proposed. In 2002, a new hazardous solid waste treatment facility was established in Jebel Ali. A year later, at the same location, a faculty that converted hazardous medical waste into non-hazardous waste was established with almost USD 1,000,000. Several laws and rules were applied to private companies to collect and transport toxic waste to the Jabel Ali treatment facility [31]. Moreover, the vision of the UAE in the waste management aspect is to recycle almost 75% of the municipal solid waste generated [32].

### 2.8. Lebanon

The main challenges faced by SWM in Lebanon include the lack of safe disposal sites, the poor efficiency in the treatment of solid waste, the lack of accountability and reliable datasets, and poor municipal, regional, and national cooperation. Many diseases have appeared due to solid waste, such as asthma, heart disease, cough, and chronic obstructive pulmonary disease. There are several proposed solutions to SWM challenges, such as developing an integrated approach entailing separation at source and collection sites and resource recovery, establishing public awareness campaigns to encourage sustainable waste management techniques that respect people's right to health and a clean environment, and highlighting the dangers of open dumping and burning. Lebanon's authorities should establish an integrated waste management system that prioritizes waste reduction and material recovery while reducing the country's dependency on landfills. The steps of recycling in Lebanon are first separating recyclables from non-recyclable garbage and then gathering recyclables for shipment to centralized sites. After that, recyclables are stored and transported to processors or remanufacturers. Finally, recyclable waste is processed to make it easier to ship or to prepare for remanufacturing.

### 2.9. Oman

Oman faces several challenges: inadequate strategy, no clear-cut master plan, a lack of specific laws and regulations, and no integrated system or facilities for effective waste [33]. There are proposed solutions for the above challenges, which are establishing improved waste management facilities with regard to waste collection from homes or away from homes and the dissemination of appropriate information and education throughout the Sultanate on waste problems. The method of waste reuse is applied in several ways. The first one is that Be'ah cooperates with Sultan Qaboos University and establishes reuse centers so that old materials are brought in to be renewed and then sold; the second way is reselling used products online.

### 2.10. Palestine

Waste can have a negative impact in Palestine, as there are many diseases caused by waste, and it can cause high cases of birth defects, infant mortality, blood diseases, organ dysfunction, and abnormalities in the immune system [5]. Climate and air pollution are also affecting many ecosystems. Landfills release methane, an extremely powerful greenhouse gas linked to climate change. Waste increases global warming in Palestine through the decomposition of organic waste, resulting in carbon dioxide and methane, which are considered greenhouse gases that contribute to global warming and climate change. Waste management also affects finance management by increasing resource security and creating new jobs. Among the main challenges in waste management are the lack of standards in the composting process, the need to develop a market for locally produced compost, and the adaptation of appropriate technologies. Waste is reused in Palestine by recycling glass, plastic, and paper/cardboard, thus producing raw materials for local industry. Solid waste is recycled, treated, stored, transported, collected, recycled, and then finally disposed of.

### 2.11. Kuwait

Kuwait's landfill sites are notorious for causing serious environmental and public health problems. Along with piling up tons of garbage, landfills also generate toxic gases (methane, carbon dioxide, etc.) and are prone to spontaneous fires [34]. The Kuwaiti

Ministry of Health showed that solid waste can cause many diseases unless it is handled properly, including respiratory and skin diseases, in addition to cancer, and it is a quick way to transmit various types of viral and bacterial infections such as hepatitis C [35]. Waste is a suitable environment for harmful insects to breed, such as mosquitoes and flies; they also release toxins into the atmosphere in addition to transmitting epidemic diseases among pets, which in turn will transmit them to humans [36]. Because of the dangers posed by the waste, Kuwait today buries waste in landfills. The area of Kuwait is 17,820 km$^2$, and 45 km$^2$ of it is landfills. This area of landfills is expected to increase to 60 km$^2$ by 2025. Since 2019, a company called Eco Star has recycled over 3.5 tonnes of plastic, 10 tonnes of paper, and 120 tonnes of metal [37]. For recycling in Kuwait, Eco Star recycles waste in the following steps [38]:

(1) Transportation and assembly: Waste is transported and collected in a safe manner in accordance with the rules of the Pollution Control Board by an experienced, trained team.
(2) Storage: The storage site has an impermeable surface and a cover that is resistant to weather factors, and wastes are placed in separate places according to their types and dangers.
(3) Separation: They have many ways of separating materials, such as using a large magnet and applying a mechanical treatment to separate metals. As for separating glass and plastic, they use water separation technology, and they also check and separate manually for better results.
(4) Recycling: The materials are prepared and recycled for sale as reusable raw materials.

*2.12. Qatar*

Waste has a negative impact on the environment and the population in Qatar. Waste causes various human diseases such as jaundice, hepatitis, gastrointestinal diseases, and others. As for the environment, the impact of waste is manifested in water pollution, harsh weather caused by climate change, air pollution, harm to animal and marine life, and human harm. Carbon dioxide and methane are two of the greenhouse gases that contribute to global warming and climate change. SWM represents one of the most pressing challenges facing Qatar due to its high population growth rate, urbanization, industrial growth, and economic expansion. The proposed solution to the challenges is the development of sustainable solutions for household solid waste by applying the principles of recycling. In 2019, the Ministry of Municipality and Al Meera Consumer Goods Company launched an initiative to recycle used batteries in order to prevent the dangerous metals that they may contain from harming the environment. Other solutions are reducing waste from landfills, recycling through collection and processing, and finally, buying new products made from recycled materials. SWM also affects finance management by increasing resource security and creating new jobs.

*2.13. Bahrain*

There are two main challenges facing Bahrain. There are no proper hazardous waste management practices, and there is a lack of regulation requiring the separation of recyclables to decrease waste. The SWM industry came up with some solutions that can be taken into consideration to solve these problems. For the hazardous waste problem, it was proposed to increase the efficiency of data collection on waste and implement a better management system. Furthermore, establishing laws to limit waste's environmental effects by sorting recyclable items would reduce the size of the problem [39,40]. The culture of reusing solid waste is only considered by waste facilities which decompose organic waste. Recycling is much more widespread in the country. Recycling facilities collect recyclables, sort them, and then manufacture them; paper makes up 13% of recyclables, plastic 7%, and glass 4% [41,42].

### 3. Applications of SWM Practices in Construction

Today, various solid wastes can be reused. As a result, in the contemporary world where natural resources have run out, evaluating construction wastes, which are employed as key aggregate sources in the construction industry, has become important. Construction waste materials are a significant environmental issue posing a risk to the environment. It is critical to both reuse and properly dispose of these materials. The construction industry can make use of waste. A summary of construction applications of SWM in different Arab countries follows.

#### 3.1. Egypt

Due to the high amounts of sand and concrete waste, recycled aggregate could be used in concrete production. Researchers have shown that the characteristics of the concrete produced with recycled aggregate are suitable for construction in Egypt. Moreover, the use of silica fumes as a mineral admixture can also enhance the performance of recycled concrete. Figure 1 shows the amount of construction and demolition waste generated in greater Cairo, and the figure shows that wood or timber is the most generated construction waste, followed by sand. Moreover, other researchers show that tires, packaging, and electronic waste are also highly produced construction waste in Egypt. Currently, there have been plans to use recycled material in the construction of pavements and tiles. There have also been exercises of reusing agricultural waste for the production of bioplastic, textiles, and paper product insulators [9]. Generally, recycling construction materials still needs to be explored in order to reuse the high amounts of construction and demolition waste in Egypt.

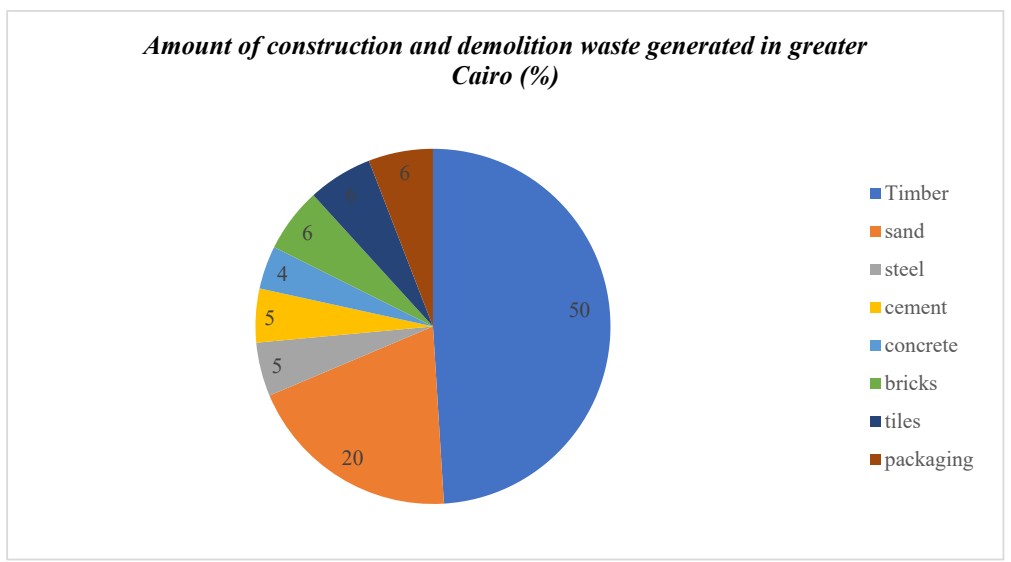

**Figure 1.** The amount of construction and demolition waste generated in greater Cairo.

#### 3.2. Iraq

In the engineering sector, the best practice is to use recycled aggregate from construction waste sites in new concrete [43]. Iraq generates about 5,859,345 tons of construction waste every year [44]. The main types of construction waste materials are concrete, metals, and wood.

#### 3.3. Saudi Arabia

In Saudi Arabia, it was discovered that just 13.6% of C&D waste is recycled and repurposed each year, with the remaining 86.4% ending up in landfills. The majority of the country's C&D waste represents a prospective supply of potentially recyclable construction materials such as debris gravel, metals, and sand. This will not only meet the KSA's gravel

and metal manufacturing needs but also handle waste disposal difficulties while providing significant economic benefits. However, in order to achieve the objective of sustainable building waste management, it is necessary to highlight the many elements that may have an impact on the country's construction waste management practices [45].

### 3.4. Yemen

Due to the relatively large amount of scrap tires wasted in Yemen, there is potential to use these scrap tires as fuel for cement kilns, as they may meet 14% of cement kilns' energy requirements [26]. Generally, the types of construction waste produced in Yemen contain sand, concrete, gravel, metal, glass, and wood. However, no data regarding any applications in the civil engineering aspect have been found, except for the collection of wood waste for cooking and fuel by local people.

### 3.5. Syria

Since the start of the Syrian crisis in 2011, solid waste treatment and transportation services have been disrupted across the country, but there is an experimental project underway to reconstruct damaged towns in a cost-effective, time-efficient, and environmentally friendly manner. According to a Syrian study, there would be 142.5 million tons of concrete and 6.65 million tons of steel as debris, in addition to metals and plastics, and approximately 0.78 million tons of solid waste [46].

### 3.6. Jordan

There are different types of construction materials in Jordan (with their range of actual waste percentage), such as concrete (2–12%), steel (2–10%), formwork (10–40%), sand, and aggregates (3–15%), cement (3–20%), bricks (5–10%), stone (5–20%), tiles (3–11%), ceramic (3–11%), pipes (3–7%), and paint (3–7%).

### 3.7. United Arab Emirates

There are several potential uses of solid waste produced in construction material. For example, fine recycled aggregates that can be obtained from construction and demolition waste can be used in the production of concrete. Figure 2 shows the percentage of construction material found in solid waste, showing that packaging materials are the highest percentage, followed by wood in the second place [7]. Other statistics show that metals, glass, aluminum, electronic devices, and tires are construction solid waste materials produced in considerable amounts in the UAE. In Dubai, tires have been used as fuel in the generation of energy as part of some industrial energy-saving plans. In 2007, almost 27.7 million tons of construction waste was dumped in the landfills, and in the first half of 2008, the generation rate of construction waste in Dubai reached 35,000 tons daily. This makes the UAE one of the biggest producers of construction waste, with almost 75% of the general solid waste generation amount.

### 3.8. Lebanon

Although construction waste accounts for the majority of Lebanon's solid waste, no management strategy or rules governing its disposal exist. Part of the generated construction waste is reused in applications such as backfilling and port expansion. Steel, for example, is being reclaimed in Lebanon for reuse or recycling [47]. The majority of construction waste, on the other hand, is deposited illegally in abandoned quarries or valleys. There are many types of solid construction waste, such as steel, concrete, glass, copper, tiles, wood, plumbing fixtures, rubble, and hazardous materials. Rubble (concrete and tiles) is more widely used and more prevalent as a type of solid construction waste.

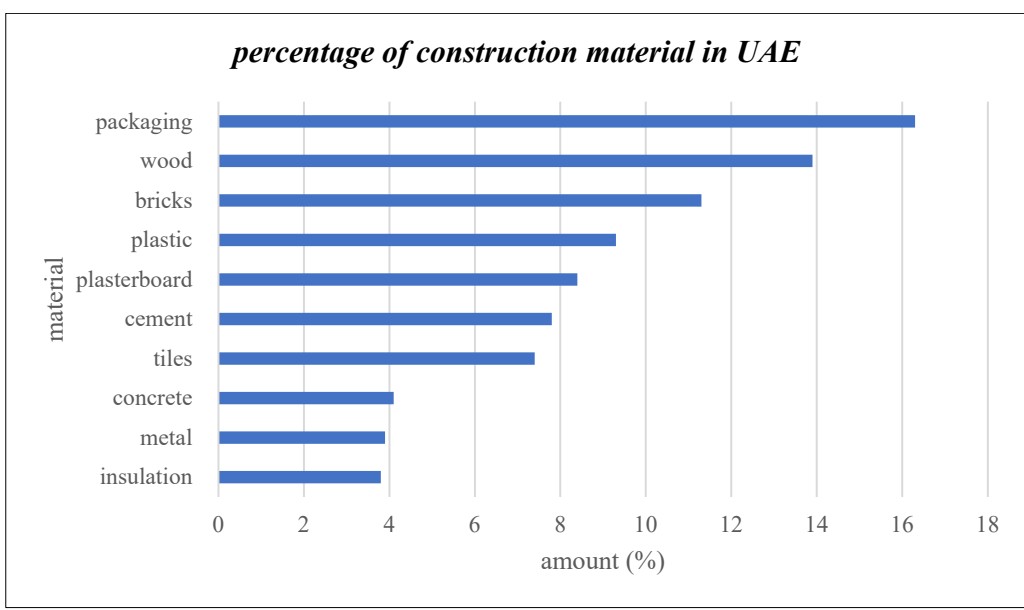

**Figure 2.** Percentage of construction materials used in the UAE.

### 3.9. Oman

Construction waste occupies a large proportion in Oman. The types of soil waste construction materials in Oman are wood, steel, concrete, and cement. Gravel, sand, and materials utilized in various industrial and civil operations, such as cement and bricks, are produced from the construction materials that were used (as waste) [48]. Oman generates about 3–5 million tons of construction waste yearly [49].

### 3.10. Palestine

In Palestine, the potential uses of solid waste construction materials are usually seen in the construction of homes, office buildings, and schools. There are many different types of solid waste construction materials for each country. For example, in Palestine, the materials include lumber, miscellaneous metal parts, packaging materials, cans, boxes, and wire. Moreover, there are many uses for solid waste from construction materials in real construction, such as in the construction of homes, office buildings, and schools, which can be seen in Palestine.

### 3.11. Kuwait

Figure 3 shows the different types of construction materials in Kuwait (with their percentage), such as concrete (30%), steel (5%), wood (8%), sand (25%), bricks (30%), and others (2%). Construction waste is usually generated in Kuwait from the demolition of old buildings, leftover from new construction, building repair and maintenance, and manufacturing debris. Based on these statistics and assumptions, the total construction waste production is estimated at 1.6 million tons/year (excluding only earth and sand), and it can be divided as follows:

(1) Demolition of buildings: 600,000 tons.
(2) Construction and repair: 600,000 tons.
(3) Construction of new buildings: 100,000 tons.
(4) Manufacturing of concrete: 300,000 tons.

Construction waste is recycled and resold in the market, and the recycled materials compete with the original materials in terms of price and quality.

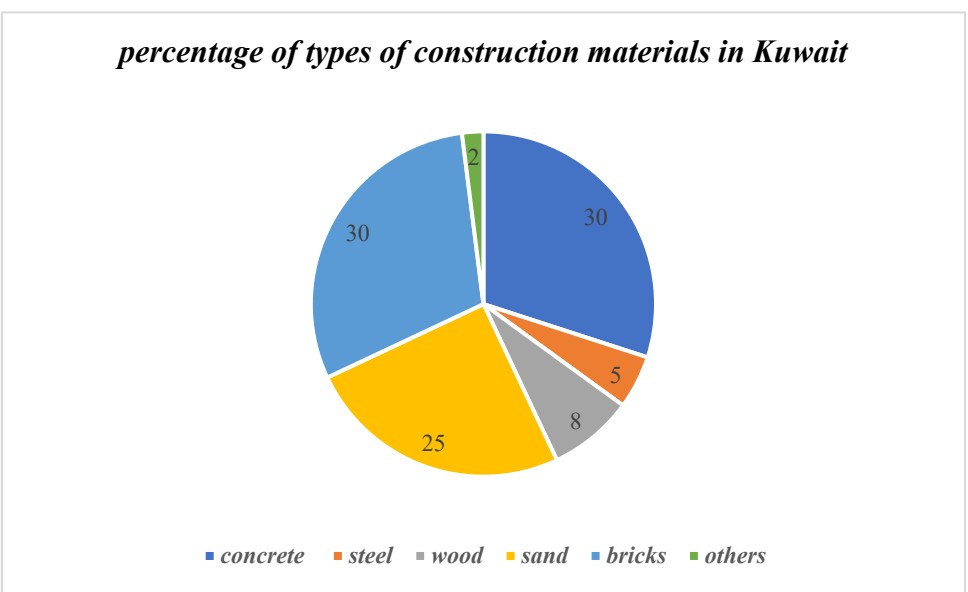

**Figure 3.** Different types of construction materials in Kuwait.

*3.12. Qatar*

The development of recycling solid waste in civil engineering has met a great need in the civil engineering field. For example, the recycled solid wastes can be used in the detoxification of wastewater and purification processes; in the production of composite machine parts from alloys of aluminum, magnesium, and copper; and the solid wastes for the asphalt production process can be modified to be reused for the improvement of frost susceptibility, moisture resistance, heat resistance, sulfate resistance, lateral and axial deformation potential, and durability potential of pavement foundations [50]. In Qatar, solid waste construction materials are mainly used in concrete buildings, soakaways, crash barriers, and concrete beams. In Qatar, concrete blocks, asphalt, steel, wood, glass, gypsum wallboard, ceramics, and roofing can be considered as types of solid waste construction materials. In addition, in Qatar, solid construction waste can be reused in parking lots, banks in canal structures, and runways of airports.

*3.13. Bahrain*

In Bahrain, there are plans to use construction waste in road construction and asphalt enhancers. The types of materials used from construction waste are concrete, bricks, glass, steel, and plastic [51]. In real construction applications, carbon dust can be used as an alternative fuel for the cement sector. The amount of construction waste generated each year at the Asker landfill is 355,690 tons [52].

## 4. Discussion and Recommendations

SWM issues in Middle Eastern Arab countries are major concerns for governments and citizens. These issues are mainly caused by several factors, including the rapid increase in the population growth rate of Arab countries as well as the alarmingly low level of social awareness among Arab societies. Figure 4 shows the population of each country along with the solid waste generation rate per year, while Figure 5 shows the population of each country along with the gross domestic product (GDP) per capita. It is observed that the solid waste generation rate is currently increasing in Arab countries mainly due to food wasting habits that are caused by the income level of a country. Social status causes certain habits and social behaviors that lead to an increase in the daily waste produced per capita.

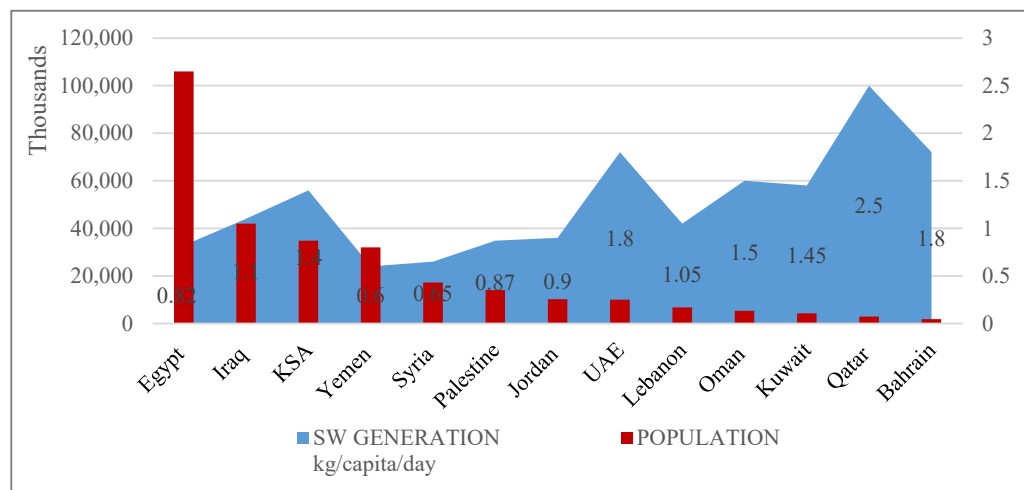

**Figure 4.** Population and waste generated in kg/capita/day for each Arab country.

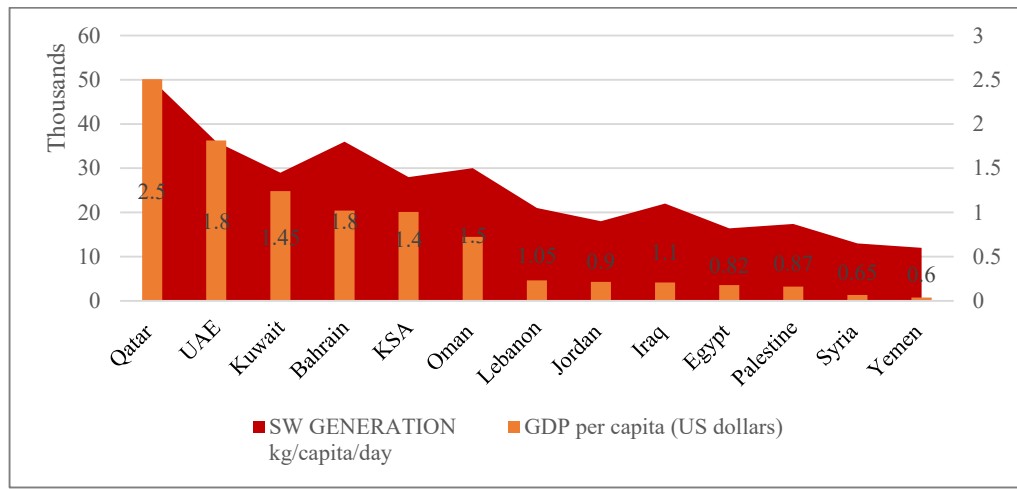

**Figure 5.** Waste generated in kg/capita/day and the GDP in US dollars/capita for each Arab country.

Several challenges are faced by Arab countries as a consequence of the increasing SWM issue. Figures 6 and 7 show the number of Arab countries that face certain challenges. It can be observed that air, soil, and water pollution are the most common challenges faced by the Arab countries studied in this review. The improper construction of sanitary landfills leads to leakage of leachate into the groundwater which affects the groundwater quality and leads to the contamination of vegetation and soil. The danger of this issue intensifies in cases of high rainfall intensity, due to the fact that this contaminated water is spread around when rain falls and rainwater flow starts. Most landfills in Arab countries are not selected based on any engineering parameters that take into consideration the hydrological and topographical suitability of the location, which leads to negative effects on the environment and water resources. These non-sanitary landfills also emit toxic gases, including methane gas and carbon dioxide gas, which are greenhouse gases that are the main causes of climate change, increasing global warming as well as threatening human, animal, and marine life. In some countries, open-air burning is a disposal method of solid waste. This leads to the emitting of dangerous gases that lead to air pollution which later on causes several lung diseases.

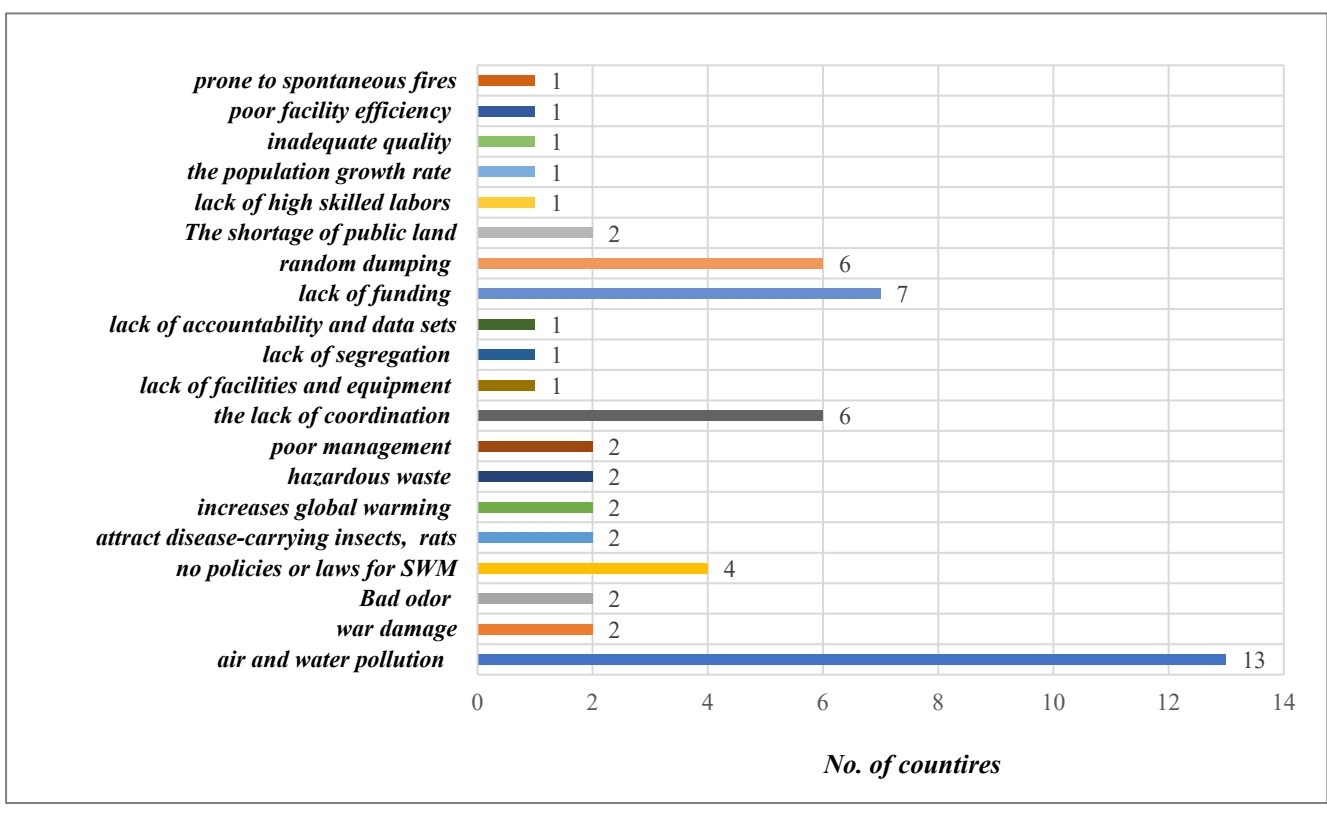

**Figure 6.** SWM-related challenges faced by Arab countries.

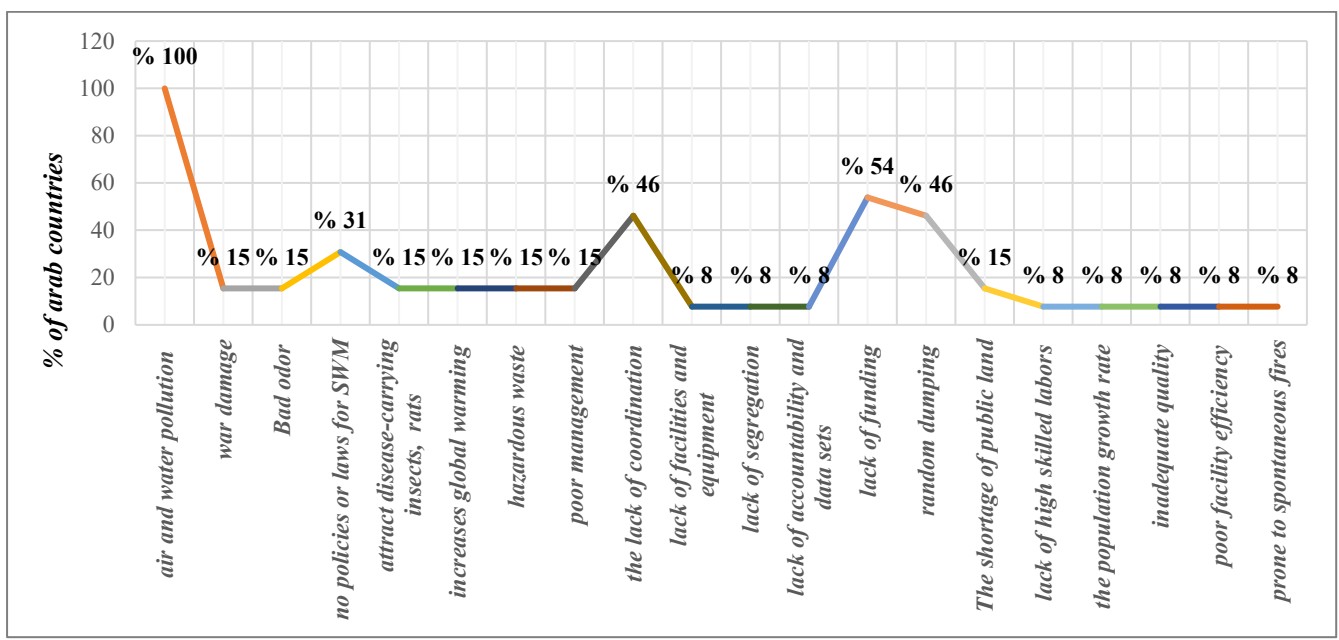

**Figure 7.** SWM-related challenges faced by Arab countries as percentages.

It was found that the second most common challenge is the lack of funding, showing that SWM should be high on governments' agendas when assigning budgets. The lack of funding leads to a lack of proper SWM systems and a lack of sufficient and effective programs that facilitate the collection, transportation, treatment, and recycling processes. Moreover, the random dumping of solid waste, as well as the lack of coordination between the government and the private sector are challenges that are faced by almost half of the Arab countries. Random dumping is a result of an ineffective collection and transportation

system, an inadequate number of landfills in the country, as well as insufficient capacity of the containers placed around the streets of the country that are supposed to hold all solid waste. Random dumping is not just an issue for health and the environment, but it also causes an aesthetic issue and results in bad odors, which cause a bad reputation that negatively affect the tourism sector. There is a noticeable lack of coordination between the governments and private companies in Middle Eastern Arab countries; governments should organize work and assign roles as well as take advantage of the private companies to help fund effective programs and plans. There is also a lack of policies and regulations that manage the attitude of citizens, as well as strict rules for industries and companies that dispose of hazardous waste in a way that negatively affects people's health.

War damage has been found to be a challenge in Iraq and Syria (Table 1). War damage leads to construction waste as well as a general state of instability in the country that creates an environment where no rules are followed and chaos ensues. Furthermore, the hazardous waste issue arises in countries such as Syria and Bahrain, meaning that these two countries have higher percentages of medical waste, chemical waste, and industrial waste that need proper facilities to treat this waste and convert it from hazardous waste to nonhazardous waste in order to be able to dispose it in sanitary landfills.

From Table 1, it can be observed that the country with the highest number of challenges is Egypt, with eight challenges, followed by Lebanon, with six challenges. This directly translates to the current state of these two countries, where solid waste is randomly dumped in unsuitable places around the streets. On the other hand, the country with the lowest number of challenges is the UAE, which is a direct reflection of the effectiveness of the plans and programs implemented. It was also observed that the UAE did not face a lack of funding, meaning that proper funding directly increases the effectiveness of SWM programs.

Table 2 shows the data acquired regarding common diseases that are caused by solid waste in Arab countries. It can be observed that hepatitis is a common disease found in Egypt, Yemen, and Qatar. Hepatitis is a medical term that describes the inflammation of the liver, where the human body's immunity system starts attacking the liver, which causes liver damage. The usual symptoms of this disease are general fatigue, abdominal pain, and nausea. Different types of infections were also found in several countries, as well as gastrointestinal diseases and cancer. All of these diseases are considered dangerous diseases that may lead to lives lost. They may also lead to a crisis if they spread around society.

Table 3 shows the construction waste applications in the civil engineering industry in Middle Eastern Arab countries. Common applications include the production of recycled concrete, enhancements in cement production, as well as asphalt production. There is a noticeably high number of applications in Palestine and Qatar, where there are civil engineering applications in the production of machine parts, asphalt production, runways of airports, pavement construction, as well as canal structures. Reusing construction waste in Arab countries has high potential since there are various studies on the effectiveness of including waste material in certain construction applications.

**Table 1.** SWM-related challenges faced by Arab countries [7–12,22,26,27,33,39,53].

| Country | Challenges | | | | | | | | |
|---|---|---|---|---|---|---|---|---|---|
| Egypt | Random dumping | The shortage of public land | Lack of skilled laborers | Lack of funding | Lack of facilities and equipment | Lack of co-ordination | Poor man-agement | Air, soil, and water pollution | 8 |
| Iraq | Random dumping | Population growth rate | Lack of funding | Air, soil, and water pollution | War damage | | | | 5 |
| KSA | Random dumping | Air, soil, and water pollution | Bad odor | Disease-carrying insects and rats | | | | | 4 |
| Yemen | Random dumping | Lack of funding | The lack of coordi-nation | Air, soil, and water pollution | No policies or laws for SWM | | | | 5 |
| Syria | Random dumping | Lack of waste seg-regation | Air, soil, and water pollution | War damage | Hazardous waste | | | | 5 |
| Palestine | The lack of coordi-nation | Air, soil, and water pollution | No policies or laws for SWM | Increases global warming | | | | | 4 |
| Jordan | The shortage of public land | Lack of funding | Air, soil, and water pollution | Bad odor | | | | | 4 |
| UAE | Inadequate quality | Air, soil, and water pollution | | | | | | | 2 |
| Lebanon | Poor facility efficiency | Random dumping | Lack of funding | Lack of ac-countability and datasets | Lack of co-ordination | Air, soil, and water pollution | | | 6 |
| Oman | Poor man-agement | Air, soil, and water pollution | No policies or laws for SWM | | | | | | 3 |
| Kuwait | Prone to sponta-neous fires | Lack of funding | Lack of coordina-tion | Air, soil, and water pollution | Disease-carrying insects and rats | | | | 5 |
| Qatar | Lack of funding | Lack of coordina-tion | Air, soil, and water pollution | Increases global warming | | | | | 4 |
| Bahrain | Air, soil, and water pollution | No policies or laws for SWM | Hazardous waste | | | | | | 3 |

**Table 2.** Diseases caused by SW in Arab countries [16,17,24–27,29,35,36].

| Country | Diseases | | | |
|---|---|---|---|---|
| Egypt | hepatitis B virus (HBV) infection | respiratory infections | gastrointestinal infections | skin infections |
| Yemen | AIDS | hepatitis | tuberculosis | |
| Palestine | birth defects | infant mortality | blood diseases | organ dysfunction | abnormalities in the immune system |
| UAE | neurological diseases | cancer | | |
| Lebanon | asthma | heart disease | cough | chronic obstructive pulmonary disease |
| Qatar | jaundice | hepatitis | gastrointestinal diseases | |

**Table 3.** Construction waste applications in Arab countries [33,34,37,41,43,45,47,48,50,51,54–56].

| Country | C&D Applications |
|---|---|
| Egypt | use of recycled material in the construction of pavements and tiles |
| Iraq | - |
| KSA | - |
| Yemen | no data regarding any applications in the civil engineering aspect have been found |
| Syria | reconstruct damaged towns in a cost-effective, time-efficient, and environmentally friendly manner |
| Palestine | detoxification of wastewater and purification processes; in the production of composite machine parts from alloys of aluminum, magnesium, and copper; and the solid wastes for asphalt production |
| Jordan | - |
| UAE | tires have been used as fuel in the generation of energy |
| Lebanon | reused in applications such as backfilling, port expansion |
| Oman | waste in recycled cement and bricks |
| Kuwait | - |
| Qatar | use of construction waste in concrete buildings, soakaways, crash barriers, concrete beams, parking lots, banks in canal structures, and runways of airports |
| Bahrain | use of construction waste in road construction and asphalt enhancers; carbon dust can be used as an alternative fuel for the cement sector |

In comparison to the high generation rates of municipal solid waste in Middle Eastern Arab countries, the actual active plans and projects are scarce. Most projects over the past few years were not sustainable and usually stopped the moment funding stopped. There is a lack of attention on the SWM system, as little to no funds are budgeted to this sector every year, leading to a lack of equipment, technology, and monitoring of the performance of these projects. Despite that, there have been successful examples in several Arab countries, which include initiating anaerobic digestion technologies in Saudi Arabia that take advantage of the high percentage of organic matter in solid waste in order to produce compost that helps fertilize soil. Moreover, in the United Arab Emirates, hazardous solid waste treatment facilities were established in Jebel Ali in 2002. This facility converts hazardous medical waste into nonhazardous waste, which helps protect the environment and people's health. UAE's vision for 2030 also aims to increase the recycling percentage to 75%. In Oman, the private company Be'ah cooperated with Sultan Qaboos University and established reuse centers to which waste materials are brought to be renewed and then sold [57,58]. In Kuwait, the current landfill area is 45 km$^2$, and there is a plan to increase this area to 60 km$^2$ by 2025. In addition, a private company in Kuwait called Eco Star recycled almost 3.5 tons of plastic, 120 tons of metal, and 10 tons of paper in 2019. Finally, in Qatar, there was a project launched by the Ministry of Municipality and Al Meera Consumer Goods Company that aimed to recycle used batteries since they are known to be a form of hazardous waste that is dangerous to the environment [59].

It was found that anaerobic digestion systems are a suitable solution for organic solid waste in Arab countries. To achieve anaerobic digestion, organic waste should be

compacted in landfills that are designed based on engineering standards that prevent leachate from leaking and contaminating the soil and the groundwater below it. Moreover, compost production has high potential in Arab countries due to the high percentage of biodegradable matter; however, there is a need to develop a market for this locally produced compost in order to use it for agricultural purposes, which could lead to a valuable financial return for the country [60,61].

Energy production and fuel production using solid waste is an industry that is yet to be explored in Arab countries. Using the appropriate technologies and equipment, energy could be recovered from solid waste and used to provide electricity to cities. Moreover, separation at the source and resource recovery should be applied. Recycling rates and applications were found to be low in Middle Eastern Arab countries, so governments should have clear plans for increasing the recycling rates in their countries.

Furthermore, there should be proper regulation laws by governments that control the random disposal of solid waste around the streets to decrease the solid waste effects on the environment and public health, as well as control the social behaviors that cause the high daily generation rate of solid waste. Moreover, the government should cooperate with the private sector and private companies in order to initiate programs and plans that contribute to the SWM sector. These plans and projects have the potential to create new jobs and help the economy of the country. The low level of social awareness regarding the issue can be improved by preparing efficient awareness-raising programs, social media campaigns, conferences, and training programs, as well as creating plans with educational institutions such as schools and universities in order to encourage sustainable waste management behaviors. It should be known to all that the solid waste issue is our responsibility as a whole and we all play a role in the system in order to achieve SWM goals in our countries.

## 5. Conclusions

In MENA Arab countries, many attempts have been made to resolve SWM issues. However, there is no sustainable effective system yet, and the health and environmental hazards caused by solid waste remain a burden on all municipalities. The main issue is that municipalities treat solid waste as a problem instead of a resource that has high potential if proper waste segregation, recycling, and reusing processes are initiated. It was also found that the social factor plays a significant role in the management process in Arab communities. It is essential to normalize minimization, waste segregation, and recycling activities in every household. It was also found that some municipalities are addressing the issue and are in the developing phase but are yet to reach the optimum system for SWM. The main challenges and issues of SW in Arab countries have not been reviewed efficiently so far. Hence, this study gives a comprehensive review of all the negative effects of SW and highlights the main struggles facing each country and the most common issues. The research recommends the concept of circular economy in terms of addressing the potential of reusing massive amounts of solid waste in sustainable construction activities, highlighting the main sectors suitable for each Arab country. Overall, the clear, comprehensive study of solid waste is key and the main step towards solving environmental and economic issues and achieving sustainable countries.

**Author Contributions:** Conceptualization, O.I.; methodology, O.I. and M.U.Q.; investigation, O.I.; resources, M.U.Q. and G.A.-K.; writing—original draft preparation, S.A.M.; writing—review and editing, M.U.Q., O.I. and G.A.-K.; supervision, O.I.; project administration, O.I.; funding acquisition, G.A.-K. All authors have read and agreed to the published version of the manuscript.

**Funding:** The research receive no external funding.

**Data Availability Statement:** Not applicable.

**Conflicts of Interest:** The authors declare no conflict of interest.

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
