# Peer review of "Challenges and Construction Applications of Solid Waste Management in Middle East Arab Countries"

_processes, doi:10.3390/pr10112289_

Round 1

Reviewer 1 Report

I have reviewed the paper ‘Challenges and Construction Applications of Solid Waste Management in Middle East Arab Countries. Overall, this paper is poorly written and hard to follow. The topic of discussion is too broad, causing the explanation to be incomplete. Some of my comments:

1. The selection of 13 Middle East Arab Countries is based on what? population or Gross domestic production (GDP)?

 2. There is no data on the amount of waste produced by 13 Middle East Arab Countries. It is suggested to obtain this data and present it as a graph (waste generation vs population).

3. There is no continuity between one verse and the next.

4. The authors touched a little bit on the functional element of solid waste (collection, treatment, disposal, and recycling) however, it would be more interesting if the author provided more detailed information (in table form) discussing how waste is managed in these 13 countries.

5. It is suggested subtopic 2 change to issues and challenges as mentioned in the abstract.

 6. Description this review paper is not the same as the aim mentioned in the abstract

7.  No citation for Tables 4 and 5. The data from Tables 4 and 5 were obtained by the author himself. If yes, so this paper is not a review paper but a technical paper. The entire paper needs to be revised.

Author Response

All comments are reviewed and sent in the uploaded version

Reviewer 2 Report

The introduction is too much generic, and more specific problems and questions (key factors to successful management?) need to analyze and should be more clearly stated. 

Author Response

All comments are reviewed and sent in the new uploaded versiin

Reviewer 3 Report

Ibrahim et al. investigated the challenges and construction applications of solid waste management in middle east Arab countries. The manuscript is innovative to some extent, but there are the following problems that need to be revised:

1 The Abstract of Part 1 is not specific and detailed enough.

2 Whether the selection of key words is appropriate.

3 Part of the introduction is not attractive.

4 The language of the manuscript needs to be greatly improved.

5 The definition of the picture in the manuscript is too low.

6 The format of references needs to be unified.

Author Response

All comments are considered and updated in the new version

Reviewer 4 Report

The authors presented research titled “Challenges and Construction Applications of Solid Waste Management in Middle East Arab”. The figures can be improved. However, the methodology of this research needs to be improved and detailed. Presented English need to be improved. Please follow the instructions below to improve the manuscript.  

1.       Last line of Abstract: what does it mean by ‘MENA’? Please use the full form of any abbreviation prior to using it. Please follow through the manuscript.

2.       Main contribution is not clear in the abstract. Please make it clear.

3.       Keywords are not enough and meaningful. Please update it accordingly.

4.       Please double-check whether the reference format is appropriate by MDPI or not.

5.       Figures are not clear. Please redraw all the figures except Fig 4 and 5.

6.       Tables are not in a good format. Please update all the tables.

7.       Discussion and Recommendations sections are good.

8.   Can you please add a summary or conclusion highlighting your key contribution and the research gaps?  

9.       Abstract, Conclusion: Please highlight the main contribution of this research.

10. Please add a comparative table or data of the presented research work with other work to better understand the reader. 

Author Response

All comments are taken in consider

Round 2

Reviewer 1 Report

I have reviewed this review paper once again and found out that this article has been improved, but most of the comments given before have not been corrected. Here are some of my comments. 

1. The last word in the abstract is the repetition of the aim of the study. Suggested the last sentences contribution of the review.

2. The selection of 13 Middle East Arab Countries is based on what? Population or Gross domestic production (GDP)?

3. There is no data on the amount of waste produced by 13 Middle East Arab Countries. It is suggested to obtain this data and present it as a graph (waste generation vs. population).

4. The last two paragraphs in the introduction are not the same as the aim of the study in the abstract.

5. It is suggested subtopic two(2) change to issues and challenges as mentioned in the abstract.

6. No references for Tables 3 and 4. Are the data in table 3 and Table 4 obtained by the author? If yes, this paper is not a review paper but a technical one. The entire article needs to be revised.

7. Use uniform terms Solid Risk management? Solid waste management? SMW?

Author Response

The comments is already updated and uploaded in the attached file

Reviewer 2 Report

questions have been all well addressed. No further comments from the reviewer.

Author Response

It is updated and attached

Reviewer 3 Report

Accept in present form

Author Response

It is updated and attached

Reviewer 4 Report

Nicely improved and can be accepted in its current form. 

Author Response

Iti s updated and attached
